# PROBABILISTIC BINARY NEURAL NETWORKS

## ABSTRACT

Low bit-width weights and activations are an effective way of combating the increasing need for both memory and compute power of Deep Neural Networks. In this work, we present a probabilistic training method for Neural Network with both binary weights and activations, called PBNet. By embracing stochasticity during training, we circumvent the need to approximate the gradient of functions for which the derivative is zero almost always, such as $\text{sign}(\cdot)$, while still obtaining a fully Binary Neural Network at test time. Moreover, it allows for anytime ensemble predictions for improved performance and uncertainty estimates by sampling from the weight distribution. Since all operations in a layer of the PBNet operate on random variables, we introduce stochastic versions of Batch Normalization and max pooling, which transfer well to a deterministic network at test time. We evaluate two related training methods for the PBNet: one in which activation distributions are propagated throughout the network, and one in which binary activations are sampled in each layer. Our experiments indicate that sampling the binary activations is an important element for stochastic training of binary Neural Networks.

## 1 INTRODUCTION

Deep Neural Networks are notorious for having vast memory and computation requirements, both during training and test/prediction time. As such, Deep Neural Networks may be unfeasible in various environments such as battery powered devices, embedded devices (because of memory requirement), on body devices (due to heat dissipation), or environments in which constrains may be imposed by a limited economical budget. Hence, there is a clear need for Neural Networks that can operate in these resource limited environments.

One method for reducing the memory and computational requirements for Neural Networks is to reduce the bit-width of the parameters and activations of the Neural Network. This can be achieved either during training (e.g., Ullrich et al. (2017); Achterhold et al. (2018)) or using post-training mechanisms (e.g., Louizos et al. (2017), Han et al. (2015)). By taking the reduction of the bit-width for weights and activations to the extreme, i.e., a single bit, one obtains a Binary Neural Network. Binary Neural Networks have several advantageous properties, i.e., a $32\times$ reduction in memory requirements and the forward pass can be implemented using XNOR operations and bit-counting, which results in a $58\times$ speedup on CPU (Rastegari et al., 2016). Moreover, Binary Neural Networks are more robust to adversarial examples (Galloway et al., 2018).

Shayer et al. (2018) introduced a probabilistic training method for Neural Networks with binary weights, but allow for full precision activations. In this paper, we propose a probabilistic training method for Neural Networks with both binary weights and binary activations, which are even more memory and computation efficient. In short, obtain a closed form forward pass for probabilistic neural networks if we constrain the input and weights to binary (random) variables. The output of the Multiply and Accumulate (MAC) operations, or pre-activation, is approximated using a factorized Normal distribution. Subsequently, we introduce stochastic versions of Max-Pooling and Batch Normalization that allow us to propagate the pre-activatoins throughout a single layer. By applying the $\text{sign}(\cdot)$ activation function to the random pre-activation, we not only obtain a distribution over binary activations, it also allows for backpropagation through the $\text{sign}(\cdot)$ operation. This is especially convenient as this in a deterministic Neural Network all gradient information is zeroed out when using sign as activation. We explore two different methods for training this probabilistic binary neural network: In the first method the activation distribution of layer $l$ is propagated to layer $(l+1)$, which

means the MAC operation is performed on two binary random variables. In the second method the binary activation is sampled as the last operation in a layer using the concrete relaxation Maddison et al. (2016). This can be thought of as a form of local reparametrization Kingma et al. (2015). We call the networks obtained using these methods PBNet and PBNet-S, respectively.

At test time, we obtain a single deterministic Binary Neural Network, an ensemble of Binary Neural Networks by sampling from the parameter distribution, or a Ternary Neural Network based on the Binary weight distribution. An advantage of our method is that we can take samples from the parameter distribution indefinitely—without retraining. Hence, this method allows for anytime ensemble predictions and uncertainty estimates. Note that while in this work we only consider the binary case, our method supports any discrete distribution over weights and activations.

## 2 PROBABILISTIC BINARY NEURAL NETWORK

In this section we introduce the probabilistic setting of the PBNet. Moreover, the approximation of the distribution on the pre-activations is introduced. For an explanation of the other operations in the PBNet, see Section 2.1 for the activation, Section 2.1.1 for the sampling of activations, and Section 2.2 for Pooling and Normalization.

We aim to train a probabilistic Binary Neural Network. As such, we pose a binary distribution over the weights of the network and optimize the parameters of this distribution instead of the parameters directly. This way, we obtain a distribution over parameters, but also deal with the inherent discreteness of a Binary Neural Network. Given an objective function $\mathcal{L}(\cdot)$, this approach can be thought of in terms of the variational optimization framework Staines & Barber (2012). Specifically, by optimizing the parameters of the weight distributions, we optimize a bound on the actual loss:

$$\min_{\mathbf{B}} \mathcal{L}(\mathbf{B}) \leq \mathbb{E}_{q_{\boldsymbol{\theta}}(\mathbf{B})}[\mathcal{L}(\mathbf{B})], \qquad (1)$$

where $\mathbf{B}$ are the binary weights of the network and $q_{\boldsymbol{\theta}}(\mathbf{B})$ is a distribution over the binary weights. For $q_{\boldsymbol{\theta}}(\mathbf{B})$ a slight reparametrization of the Bernoulli distribution is used, which we will refer to as the Binary distribution. This distribution is parameterized by $\theta \in [-1, 1]$ and is defined by:

$$a \sim \text{Binary}(\theta) \iff \frac{a+1}{2} \sim \text{Bernoulli}(\frac{\theta+1}{2}). \quad (2)$$

For the properties of this distribution, please refer to Appendix A.

We will now consider using the Binary distribution for both the weights and the activations in a Neural Network. Since the pre-activations in a Neural Network are computed using MAC operations, which are the same for each value in the pre-activation, we will only consider a single value in our discussion here. Let $\mathbf{w} \sim \text{Binary}(\boldsymbol{\theta})$ and $\mathbf{h} \sim \text{Binary}(\boldsymbol{\phi})$ be the weight and input random variable for a given layer. As such, the inner-product between the weights and input is distributed according to a translated and scaled Poisson binomial distribution:

$$\frac{\mathbf{w} \cdot \mathbf{h} + D}{2} \sim \text{PoiBin}(2[\boldsymbol{\theta} \odot \boldsymbol{\phi}] - 1). \quad (3)$$

---

**Algorithm 1:** Pseudo code for forward pass of single layer in PBNet(-S). $\mathbf{a}_{l-1}$ denotes the activation of the previous layer, $\mathbf{B}$ the random binary weight matrix, $\tau$ is the temperature used for the concrete distribution, $f(\cdot, \cdot)$ the linear transformation used in the layer, $\epsilon > 0$ a small constant for numerical stability, $D$ the dimensionality of the inner product in $f$, and $\gamma$ & $\beta$ are the parameters for batch normalization.

**Input:** $\mathbf{a}_{l-1}, \mathbf{B} \sim p(\mathbf{B}), \tau, f(\cdot, \cdot), \epsilon, \gamma, \beta$
**Result:** Binary activation $\mathbf{a}_l$

```
// CLT approximation
```
**if** $\mathbf{a}_{l-1}$ *is a binary random variable* **then**
   $\boldsymbol{\mu} = f(\mathbb{E}[\mathbf{B}], \mathbb{E}[\mathbf{a}_{l-1}])$;
   $\boldsymbol{\sigma}^2 = D - f((\mathbb{E}[\mathbf{B}])^2, (\mathbb{E}[\mathbf{a}_{l-1}])^2)$;
**else**
   $\boldsymbol{\mu} = f(\mathbb{E}[\mathbf{B}], \mathbf{a}_{l-1})$;
   $\boldsymbol{\sigma}^2 = f(\mathbb{V}[\mathbf{B}], \mathbf{a}_{l-1}^2)$;
**end**

```
// Batch normalization
```
$\mathbf{m} = \text{channel-wise-mean}(\boldsymbol{\mu})$;
$\mathbf{v} = \text{channel-wise-variance}(\boldsymbol{\mu}, \boldsymbol{\sigma}^2, \mathbf{m})$;
$\boldsymbol{\mu} = \gamma(\boldsymbol{\mu} - \mathbf{m})/\sqrt{\mathbf{v} + \epsilon} + \beta$;
$\boldsymbol{\sigma}^2 = \gamma^2 \boldsymbol{\sigma}^2/(\mathbf{v} + \epsilon)$;

```
// Max pooling
```
**if** *max pooling required* **then**
   $\mathbf{n} \sim \mathcal{N}(0, \mathbf{I})$;
   $\mathbf{s} = \boldsymbol{\mu} + \boldsymbol{\sigma} \odot \mathbf{n}$;
   $\boldsymbol{\iota} = \text{max-pooling-indices}(\mathbf{s})$;
   $\boldsymbol{\mu}, \boldsymbol{\sigma}^2 = \text{select-at-indices}(\boldsymbol{\mu}, \boldsymbol{\sigma}^2, \boldsymbol{\iota})$;
**end**

```
// Binarization and sampling
```
$\mathbf{p} = 1 - \Phi(0 | \boldsymbol{\mu}, \boldsymbol{\sigma}^2)$;
**if** *sample activation* **then**
   $\mathbf{a}_l \sim \text{BinaryConcrete}(\mathbf{p}, \tau)$;
   **return** $\mathbf{a}_l$;
**else**
   **return** $\text{Binary}(\mathbf{p})$
**end**

---

Where $D$ is the dimensionality of $\mathbf{h}$ and $\mathbf{w}$ and $\odot$ denotes element-wise multiplication. See the *picket fence* on the top in Figure 1 for an illustration of the PMF of a Poisson binomial distribution. Although the scaled and translated Poisson binomial distribution is the exact solution for the inner product between the weight and activation random variables, it is hard to work with in subsequent layers. For this reason, and the fact that the Poisson binomial distribution is well approximated by a Normal distribution (Wang & Manning, 2013), we use a Normal approximation to the Poisson binomial distribution, which allows for easier manipulations. Using the properties of the Binary distribution and the Poisson binomial distribution, the approximation for the pre-activation $a$ is given by:

$$a = \mathbf{w} \cdot \mathbf{h} \quad \dot\sim \quad \mathcal{N}(\sum_{d=1}^{D} \theta_d \phi_d, D - \sum_{d=1}^{D} \theta_d^2 \phi_d^2). \tag{4}$$

Note that, this is exactly the approximation one would obtain by using the Lyapunov Central Limit Theorem (CLT), which was used by Shayer et al. (2018). This allows us to obtain a close approximation to the pre-activation distribution, which we can propagate through the layer and/or network. So far, only the MAC operation in a given layer is discussed, in Section 2.1 application of the binary activation is discussed and in Section 2.1. The stochastic versions of Batch Normalization and Max Pooling are introduced in Section 2.2. For specifics on sampling the binary activation, see Section 2.1.1. The full forward pass for a single layer is given in detail in Algorithms 1.

## 2.1 STOCHASTIC BINARY ACTIVATION

Since the output of a linear operation using binary inputs is not restricted to be binary, it is required to apply a binarization operation to the pre-activation in order to obtain binary activations. Various works – e.g., Hubara et al. (2016) and Rastegari et al. (2016) – use either deterministic or stochastic binarization functions, i.e.,

$$b_{\text{det}}(a) = \begin{cases} +1 & \text{if } a \geq 0 \\ -1 & \text{otherwise} \end{cases} \quad b_{\text{stoch}}(a) = \begin{cases} +1 & \text{with probability } p = \text{sigmoid}(a) \\ -1 & \text{with probability } 1 - p \end{cases}. \tag{5}$$

In our case the pre-activations are random variables. Hence, applying a deterministic binarization function to a random pre-activations results in a stochastic binary activation. Specifically, let $a_i \sim \mathcal{N}(\mu_i, \sigma_i^2)$ be a random pre-ctivation obtained using the normal approximation, as introduced in the previous section, then the activation (after binarization) is again given as a Binary random variable". Interestingly, the Binary probability can be computed in closed form by evaluating the probability density that lies above the binarization threshold:

$$h_i = b_{\text{det}}(a_i) \sim \text{Binary}(q_i), \quad q_i = 1 - \Phi(0|\mu_i, \sigma_i^2), \tag{6}$$

where $\Phi(\cdot|\boldsymbol{\mu}, \boldsymbol{\sigma}^2)$ denotes the CDF of $\mathcal{N}(\boldsymbol{\mu}, \boldsymbol{\sigma}^2)$. Applying the binarization function to a random pre-activation has two advantages. First, the derivatives $\partial q_i / \partial \mu_i$ and $\partial q_i / \partial \sigma_i$ are not zero almost everywhere, in contrast to the derivatives of $b_{\text{det}}$ and $b_{\text{stoch}}$ when applied to a deterministic input. Second, the distribution over $h_i$ reflects the true uncertainty about the sign of the activation, given the stochastic weights, whereas $b_{\text{stoch}}$ uses the magnitude of the pre-activation as a substitute. For example, a pre-activation with a high positive magnitude and high variance will be deterministically mapped to 1 by $b_{\text{stoch}}$. In contrast, our method takes the variance into account and correctly assigns some probability mass to $-1$. See Figure 1 for a graphical depiction of the stochastic binary activation.

### 2.1.1 SAMPLING THE BINARY ACTIVATIONS

So far, we have discussed propagating distributions throughout the network. Alternatively, the binary activations can be sampled using the Concrete distribution (Maddison et al., 2016) during training. specifically, we use the *hard* sample method as discussed by Jang et al. (2016). By sampling the activations, the input for subsequent layers will match the input that is observed at test time more closely.

As a consequence of sampling the activation, the input to a layer is no longer a distribution but a $\mathbf{h} \in \{-1, +1\}^D$ vector instead. As such, the normal approximation to the pre-activation is computed

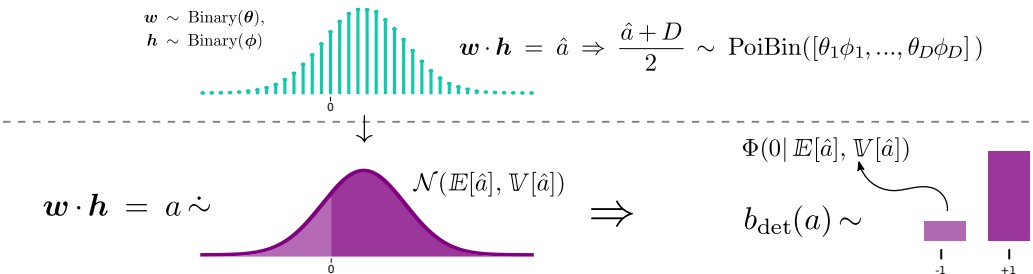

Figure 1: The discrete Poisson binomial distribution (in green) is approximated by a continuous Gaussian distribution (in purple). By applying $b_{\text{det}}$ to a random pre-activation we obtain a binary activation distribution.

slightly different. From the Lyapunov CLT it follows that the approximation to the distribution of the pre-activation is given by:

$$a = \mathbf{w} \cdot \mathbf{h} \quad \dot{\sim} \quad \mathcal{N}(\sum_{d=1}^{D} \theta_d h_d, \sum_{d=1}^{D} \theta_d^2 h_d^2), \tag{7}$$

where $\mathbf{w} \sim \text{Binary}(\boldsymbol{\theta})$ is a random weight. Similarly, the pre-activation of the input layer is also computed using this approximation—given a real-valued input vector. We will refer to a PBNet that uses activation sampling as PBNet-S.

## 2.2 Normalization and Pooling

Other than a linear operation and an (non-linear) activation function, Batch Normalization (Ioffe & Szegedy, 2015) and pooling are two popular building blocks for Convolutional Neural Networks. For Binary Neural Networks, applying Batch Normalization to a binarized activation will result in a non-binary result. Moreover, the application of max pooling on a binary activation will result in a feature map containing mostly +1s. Hence, both operations must be applied before binarization. However, in the PBNet, the binarization operation is applied before sampling. As a consequence, the Batch Normalization and pooling operations can only be applied on random pre-activations. For this reason, we define these methods for random variables. Although there are various ways to define these operation in a stochastic fashion, our guiding principle is to only leverage stochasticity during training, i.e., at test time, the stochastic operations are replaced by their conventional implementations and parameters learned in the stochastic setting must be transferred to their deterministic counterparts.

### 2.2.1 Stochastic Batch Normalization

Batch Normalization (BN) (Ioffe & Szegedy, 2015) — including an affine transformation — is defined as follows:

$$\hat{\mathbf{a}}_i = \frac{\mathbf{a}_i - \mathbf{m}}{\sqrt{\mathbf{v} + \epsilon}} \gamma + \beta, \tag{8}$$

where $\mathbf{a}_i$ denotes the pre-activation before BN, $\hat{\mathbf{a}}$ the pre-activation after BN, and $\mathbf{m}$ & $\mathbf{v}$ denote the sample mean and variance of $\{\mathbf{a}_i\}_{i=1}^{M}$, for an $M$-dimensional pre-activation. In essence, BN translates and scales the pre-activations such that they have approximately zero mean and unit variance, followed by an affine transformation. Hence, in the stochastic case, our aim is that samples from the pre-activation distribution after BN also have approximately zero mean and unit variance—to ensure that the stochastic batch normalization can be transfered to a deterministic binary neural network. This is achieved by subtracting the population mean from each pre-activation random variable and by dividing by the population variance. However, since $\mathbf{a}_i$ is a random variable in the PBNet, simply using the population mean and variance equations will result in non-standardized output. Instead, to ensure a standardized distribution over activations, we compute the expected

population mean and variance under the pre-activation distribution:

$$\mathbb{E}_{p(\mathbf{a}|\mathbf{B},\mathbf{h})}[\mathbf{m}] = \mathbb{E}\left[\frac{1}{M}\sum_{i=1}^{M}\mathbf{a}_i\right] = \frac{1}{M}\sum_{i=1}^{M}\mathbb{E}\left[\mathbf{a}_i\right] = \frac{1}{M}\sum_{i=1}^{M}\boldsymbol{\mu}_i \tag{9}$$

$$\mathbb{E}_{p(\mathbf{a}|\mathbf{B},\mathbf{h})}[\mathbf{v}] = \mathbb{E}\left[\frac{1}{M-1}\sum_{i=1}^{M}(\mathbf{a}_i - \mathbb{E}[\mathbf{m}])^2\right] = \frac{1}{M-1}\left\{\sum_{i=1}^{K}\boldsymbol{\sigma}_i^2 + \sum_{i=1}^{M}(\boldsymbol{\mu}_i - \mathbb{E}[\mathbf{m}])^2\right\}, \tag{10}$$

where $M$ is the total number of activations and $\mathbf{a}_i \sim \mathcal{N}(\boldsymbol{\mu}_i, \boldsymbol{\sigma}_i)$ are the random pre-activations. By substituting $\mathbf{m}$ and $\mathbf{v}$ in Equation 8 by Equation 9 and 10, we obtain the following batch normalized Gaussian distributions for the pre-activations:

$$\hat{\mathbf{a}}_i = \frac{\mathbf{a}_i - \mathbb{E}[\mathbf{m}]}{\sqrt{\mathbb{E}[\mathbf{v}] + \epsilon}}\gamma + \beta \quad \Rightarrow \quad \hat{\mathbf{a}}_i \sim \mathcal{N}\left(\frac{\boldsymbol{\mu}_i - \mathbb{E}[\mathbf{m}]}{\sqrt{\mathbb{E}[\mathbf{v}] + \epsilon}}\gamma + \beta, \frac{\gamma^2}{\mathbb{E}[\mathbf{v}] + \epsilon}\boldsymbol{\sigma}_i^2\right). \tag{11}$$

Note that this assumes a single channel, but is easily extended to 2d batch norm in a similar fashion as conventional Batch Normalization. At test time, Batch Normalization in a Binary Neural Network can be reduced to an addition and sign flip of the activation, see Appendix B for more details.

### 2.2.2 STOCHASTIC MAX POOLING

In general, pooling applies an aggregation operation to a set of (spatially oriented) pre-activations. Here we discuss max pooling for stochastic pre-activations, however, similar considerations apply for other types of aggregation functions.

In the case of max-pooling, given a spatial region containing stochastic pre-activations $\mathbf{a}_1, \ldots, \mathbf{a}_K$, we aim to stochastically select one of the $\mathbf{a}_i$. Note that, although the distribution of $\max(\mathbf{a}_1, \ldots, \mathbf{a}_K)$ is well-defined (Nadarajah & Kotz, 2008), its distribution is not Gaussian and thus does not match one of the input distributions. Instead, we sample one of the input random variables in every spatial region according to the probability of that variable being greater than all other variables, i.e., $\rho_i = p(\mathbf{a}_i > \mathbf{z}_{\setminus i})$, where $\mathbf{z}_{\setminus i} = \max(\{\mathbf{a}_j\}_{j \neq i})$. $\rho_i$ could be obtained by evaluating the CDF of $(\mathbf{z}_{\setminus i} - \mathbf{a}_i)$ at 0, but to our knowledge this has no analytical form. Alternatively, we can use Monte-Carlo integration to obtain $\boldsymbol{\rho}$:

$$\boldsymbol{\rho} \approx \frac{1}{L}\sum_{l=1}^{L}\text{one-hot}(\arg\max \mathbf{s}^{(l)}), \quad \mathbf{s}^{(l)} \sim p(\mathbf{a}_1, \mathbf{a}_2, \ldots, \mathbf{a}_K) = \prod_{i=1}^{K}\mathcal{N}(\boldsymbol{\mu}_i, \boldsymbol{\sigma}_i^2) \tag{12}$$

where one-hot($i$) returns a $K$-dimensional one-hot vector with the $i$th elements set to one. The pooling index $\iota$ is then sampled from $\text{Cat}(\boldsymbol{\rho})$. However, more efficiently, we can sample $\mathbf{s} \sim p(\mathbf{a}_1, \ldots, \mathbf{a}_K)$ and select the index of the maximum in $\mathbf{s}$, which is equivalent sampling from $\text{Cat}(\boldsymbol{\rho})$. Hence, for a given max pooling region, it is sufficient to obtain a single sample from each normal distribution associated with each pre-activation and *keep* the random variable for which this sample is maximum. A graphical overview of this is given in Figure 2.

Other forms of stochastic or probabilistic max pooling were introduced by Lee et al. (2009) and Zeiler & Fergus (2013), however, in both cases a single activation is sampled based on the magnitude of the activations. In contrast, in our procedure we stochastically propagate one of the input distributions over activations.

### 2.3 WEIGHT INITIALIZATION

For the PBNet the parameters $\boldsymbol{\theta}$ for $q_{\boldsymbol{\theta}}(\mathbf{B})$ are initialized from a uniform $U(-1, 1)$ distribution. Although the final parameter distribution more closely follows a $\text{Beta}(\alpha, \alpha)$ distribution, for $\alpha < 1$, we did not observe any significant impact choosing another initialization method for the PBNet.

In the case of the PBNet-S, we observed a significant improvement in training speed and performance by initializing the parameters based on the parameters of a pre-trained full precission Neural Network. This initializes the convolutional filters with more structure than a random initialization. This is desirable as in order to flip the value of a weight, the parameter governing the weight has to pass through a high variance regime, which can slow down convergence considerably.

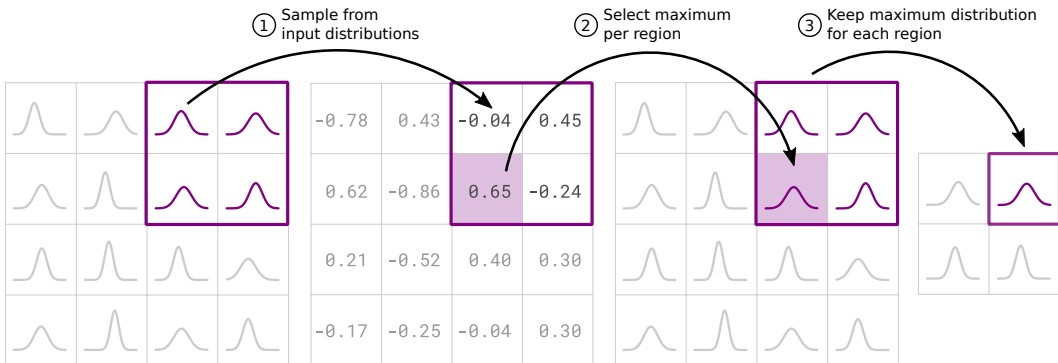

Figure 2: Max pooling for random variables is performed by taking a single sample from each of the input distributions. The output random variable for each pooling region is the random variable that is associated with the maximum sample.

For the PBNet-S, We use the weight transfer method introduced by Shayer et al. (2018) in which the parameters of the weight distribution for each layer are initialized such that the expected value of the random weights equals the full precision weight divided by the standard deviation of the weights in the given layer. Since not all rescaled weights lay in the $[-1, 1]$ range, all binary weight parameters are clipped between $[-0.9, 0.9]$. This transfer method transfers the structure present in the filters of the full precision network and ensures that a significant part of the parameter distributions is initialized with low variance.

## 2.4 Deterministic Binary Neural Network

In our training procedure, a stochastic neural network is trained. However, at test time (or on hardware) we want to leverage all the advantages of a full binary Neural Network. Therefore, we obtain a deterministic binary Neural Network from the parameter distribution $q_\theta(\mathbf{B})$ at test time. We consider three approaches for obtaining a deterministic network: a deterministic network based on the mode of $q_\theta(\mathbf{B})$ called PBNET-MAP, an ensemble of binary Neural Networks sampled from $q_\theta(\mathbf{B})$ named PBNET-$x$, and a ternary Neural Network (PBNET-TERNARY), in which a single parameter $W_i$ may be set to zero based on $q_\theta$, i.e.:

$$W_i = \begin{cases} +1 & \text{if } q_\theta(B_i = +1) \geq 3/4 \\ -1 & \text{if } q_\theta(B_i = -1) \geq 3/4 \\ 0 & \text{otherwise} \end{cases} \tag{13}$$

The ternary network can also be viewed as a sparse PBNet, however, sparse memory look-ups may slow down inference.

Note that, even when using multiple binary neural networks in an ensemble, the ensemble is still more efficient in terms of computation and memory when compared to a full precision alternative. Moreover, it allows for anytime ensemble predictions for improved performance and uncertainty estimates by sampling from the weight distribution.

Since the trained weight distribution is not fully deterministic, the sampling of individual weight instantiations will result in a shift of the batch statistics. As a consequence, the learned batch norm statistics no longer closely match the true statistics. This is alleviated by re-estimating the batch norm statistics based on (a subset of) the training set after weight sampling using a moving mean and variance estimator. We observed competitive results using as little as 20 batches from the training set.

## 3 Related Work

Binary and low precision neural networks have received significant interest in recent years. Most similar to our work, in terms of the final neural network, is the work on Binarized Neural Networks by Hubara et al. (2016). in this work a real-valued shadow weight is used and binary weights are

obtained by binarizing the shadow weights. Similarly the pre-activations are binarized using the same binarization function. In order to back-propagate through the binarization operation the straight-through estimator (Hinton, 2012) is used. Several extensions to Binarized Neural Networks have been proposed which — more or less — qualify as binary neural networks: XNOR-net (Rastegari et al., 2016) in which the real-valued parameter tensor and activation tensor is approximated by a binary tensor and a scaling factor per channel. ABC-nets Lin et al. (2017) take this approach one step further and approximate the weight tensor by a linear combination of binary tensors. Both of these approaches perform the linear operations in the forward pass using binary weights and/or binary activations, followed by a scaling or linear combination of the pre-activations. In McDonnell (2018), similar methods to Hubara et al. (2016) are used to binarize a wide resnet (Zagoruyko & Komodakis, 2016) to obtain results on ImageNet very close to the full precision performance. Another method for training binary neural networks is Expectation Backpropagation (Soudry et al., 2014) in which the central limit theorem and online expectation propagation is used to find an approximate posterior. This method is similar in spirit to ours, but the training method is completely different. Most related to our work is the work by Shayer et al. (2018) which use the local reparametrization trick to train a Neural Network with binary weights and the work by Baldassi et al. (2018) which also discuss a binary Neural Network in which the activation distribution are propagated through the network. Moreover, in (Wang & Manning, 2013) the CLT was used to approximate dropout noise during training in order to speed up training, however, there is no aim to learn binary (or discrete) weights or use binary activations in this work.

## 4 EXPERIMENTS

We evaluate the PBNet on the MNIST and CIFAR-10 benchmarks and compare the results to Binarized Neural Networks (Hubara et al., 2016), since the architectures of the deterministic networks obtained by training the PBNet are equivalent.

### 4.1 EXPERIMENTAL DETAILS

The PBNets are trained using either a cross-entropy (CE) loss or a binary cross entropy for each class (BCE). For the CE loss there is no binarization step in the final layer, instead the mean of the Gaussian approximation is used as the input to a softmax layer. For BCE, there is a binarization step, and we treat the probability of the $i$th output being $+1$ as the probability of the input belonging to the $i$th class. Specifically, for an output vector $\mathbf{p} \in [0,1]^C$ for $C$ classes and the true class $y$, the BCE loss for a single sample is defined as

$$\mathcal{L}_{\text{BCE}}(\mathbf{p}, y) = -\sum_{c=1}^{C} [c = y] \log p_c + [c \neq y] \log(1 - p_c). \tag{14}$$

The weights for the PBNet-S networks are initialized using the transfer method described in Section 2.3 and the PBNets are initialized using a uniform initialization scheme. All models are optimized using Adam (Kingma & Ba, 2014) and a validation loss plateau learning rate decay scheme. We keep the temperature for the binary concrete distribution static at 1.0 during training. For all settings, we optimize model parameters until convergence, after which the best model is selected based on a validation set. Our code is implemented using PyTorch (Paszke et al., 2017).

For Binarized Neural Networks we use the training procedure described by Hubara et al. (2016), i.e., a squared hinge loss and layer specific learning rates that are determined based on the Glorot initialization method (Glorot & Bengio, 2010).

Experimental details specific to datasets are given in Appendix C and the results are presented in Table 1. We report both test set accuracy obtained after binarizing the network as well as the the test set accuracy obtained by the stochastic network during training (i.e., by propagating activation distributions).

### 4.2 ENSEMBLE BASED UNCERTAINTY ESTIMATION

As presented in Table 1 the accuracy improves when using an ensemble. Moreover, the predictions of the ensemble members can be used to obtain an estimate of the certainty of the ensemble as a whole.

Table 1: Test accuracy on MNIST and CIFAR-10 for Binarized NN (Hubara et al., 2016), PBNet, and a full precission network (FPNet). PBNet-map refers to a deterministic PBNet using the map estimate, PBNet-Ternary is a ternary deterministic network obtained from $q_\theta$, and PBNet-$X$ refers to an ensemble of $X$ networks, each sampled from the same weight distribution. For the ensemble results both mean and standard deviation are presented. The propagate column contains results obtained using the stochastic network whereas results in the binarized column are obtained using a deterministic binary Neural Network.

| | MNIST | | CIFAR-10 | |
| --- | --- | --- | --- | --- |
| | PROPAGATE | BINARIZED | PROPAGATE | BINARIZED |
| BINARIZED NN | – | 99.17 | – | 88.17 |
| PBNET-MAP (BCE) | 99.35 | 99.13 | 88.24 | 79.98 |
| PBNET-MAP (CE) | 99.24 | 98.64 | 86.73 | 75.05 |
| PBNET-S-MAP (BCE) | 99.26 | 99.22 | 89.58 | 89.10 |
| PBNET-S-MAP (CE) | 99.14 | 99.05 | 88.67 | 88.54 |
| PBNET-S-TERNARY (BCE) | 99.26 | | 89.70 | |
| PBNET-S-2 (BCE) | $99.25 \pm 0.047$ | | $89.75 \pm 0.205$ | |
| PBNET-S-5 (BCE) | $99.29 \pm 0.036$ | | $90.75 \pm 0.202$ | |
| PBNET-S-16 (BCE) | $99.30 \pm 0.025$ | | $91.28 \pm 0.112$ | |
| FPNET | 99.48 | | 92.45 | |

To evaluate this, we plot an error-coverage curve (Geifman & El-Yaniv, 2017) in Figure 3a. This curve is obtained by sorting the samples according to a statistic and computing the error percentage in the top $x\%$ of the samples – according to the statistic. For the Binarized Neural Network and PBNet-MAP the highest softmax score is used, whereas for the ensembles the variance in the prediction of the top class is used. The figure suggests that the ensemble variance is a better estimator of network certainty, and moreover, the estimation improves as the ensemble sizes increases.

### 4.3 EFFECT OF BATCH STATISTICS RE-ESTIMATION

As discussed in Section 2.4, after sampling the parameters of a deterministic network the batch statistics used by Batch Normalization must be re-estimated. Figure 3b shows the results obtained using a various number of batches from the training set to re-estimate the statistics. This shows that even a small number of samples is sufficient to estimate the statistics.

### 4.4 ABLATION STUDIES

We perform an ablation study on both the use of (stochastic) Batch Normalization and the use of weight transfer for the PBNet-S on CIFAR-10. For Batch Normalization, we removed all batch normalization layers from the PBNet-S and retrained the model on CIFAR-10. This resulted in a test set accuracy of 79.21%. For the weight initialization experiment, the PBNet-S weights are initialized using a uniform initialization scheme and is trained on CIFAR-10, resulting in a test set accuracy of 83.61%. Moreover, the accuracy on the validation set during training is presented in Figure 3c. Note that these numbers are obtained *without* sampling a binarized network from the weight distribution, i.e., local reparametrization and binary activation samples are used. The PBNet-S that uses both weight transfer and stochastic Batch Normalization results in a significant performance improvement, indicating that both stochastic Batch Normalization and weight transfer are necessary components for the PBNet-S.

### 4.5 SAMPLING OF BINARY ACTIVATIONS

The results of our experiments show that, following our training procedure, sampling of the binary activations is a necessary component. Although the stochastic PBNet generalizes well to unseen data, there is a significant drop in test accuracy when a binary Neural Network is obtained from

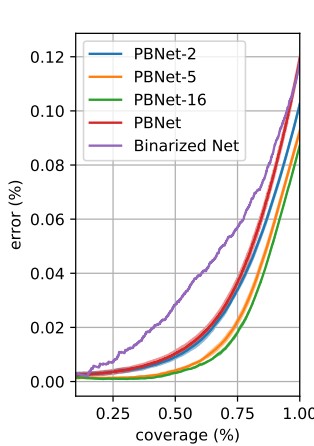 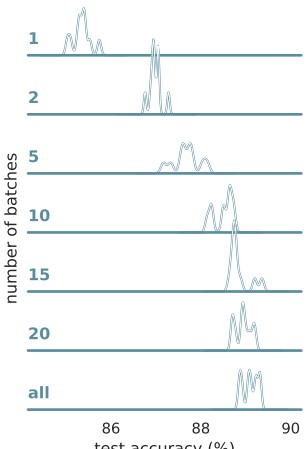 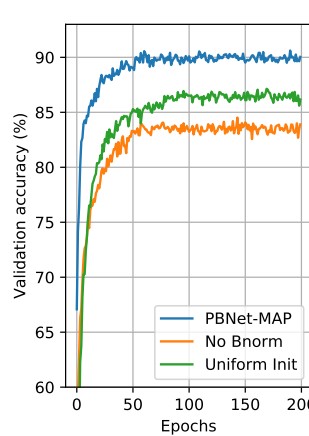

(a) Error coverage for CIFAR-10.

(b) Test set performance with increasing number of batches used to re-estimate the batch statistics on CIFAR-10.

(c) Accuracy on validation set during training, i.e., using stochastic weights, local reparametrization and binary activation sampling.

Figure 3: Error coverage curve, batch statistic re-estimation results and ablation study results for CIFAR-10.

the stochastic PBNet. In contrast, this performance drop is not observed for PBNet-S. A potential explanation of this phenomenon is that by sampling the binary activation during training, the network is forced to become more robust to the inherent binarization noise that is present at test time of the binarized Neural Network. If this is the case, then sampling the binary activation can be thought of as a regularization strategy that *prepares* the network for a more noisy binary setting. However, other regularization strategies may also exist.

## 5    CONCLUSION

We have presented a stochastic method for training Binary Neural Networks. The method is evaluated on multiple standardized benchmarks and reached competitive results. The PBNet has various advantageous properties as a result of the training method. The weight distribution allows one to generate ensembles online which results in improved accuracy and better uncertainty estimations. Moreover, the Bayesian formulation of the PBNet allows for further pruning of the network, which we leave as future work.

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

## A  BINARY DISTRIBUTION

For convenience, we have introduced the Binary distribution in this paper. In this appendix we list some of the properties used in the paper, which all follow direcly from the properties of the Bernoulli distribution. The Binary distribution is a reparametrization of the Bernoulli distribution such that:

$$a \sim \text{Binary}(\theta) \iff \frac{a+1}{2} \sim \text{Bernoulli}(\frac{\theta+1}{2}). \tag{15}$$

This gives the following probability mass function:

$$\text{Binary}(a|\theta) = \frac{1}{2}(\theta+1)^{\frac{1}{2}(a+1)}(1-\theta)^{\frac{1}{2}(1-a)} \tag{16}$$

where $a \in \{-1, +1\}$ and $\theta \in [-1, 1]$. From this, the mean and variance are easily computed:

$$\mathbb{E}[a] = \theta, \quad \mathbb{V}[a] = 1 - \theta^2. \tag{17}$$

Finally, let $b \sim \text{Binary}(\phi)$, then $ab \sim \text{Binary}(\theta\phi)$.

## B  BATCH NORMALIZATION IN A BINARY NEURAL NETWORK

During training the PBNet is trained using stochastic Batch Normalization. At test time, the parameters learned using stochastic Batch Normalization can be transferred to a *conventional* Batch Normalization implementation. Alternatively, Batch Normalization can be reduced to an (integer) addition and multiplication by $\pm 1$ after applying the sign activation function. Given a pre-activation $a$, the application of Batch Normalization followed by a $\text{sign}$ binarization function can be rewritten as:

$$\text{sign}\left(\frac{a-m}{\sqrt{v+\epsilon}}\gamma + \beta\right) = \text{sign}\left(\frac{a}{\sqrt{v+\epsilon}}\gamma - \frac{m}{\sqrt{v+\epsilon}}\gamma + \beta\right) \tag{18}$$

$$= \text{sign}\left(a\gamma - m\gamma + \beta\sqrt{v+\epsilon}\right) \tag{19}$$

$$= \text{sign}(\gamma)\,\text{sign}\left(a - m + \frac{\beta}{\gamma}\sqrt{v+\epsilon}\right) \tag{20}$$

when $a \in \mathbb{Z}$, which is the case for all but the first layer

$$= \text{sign}(\gamma)\,\text{sign}\left(a + \underbrace{\left\lfloor \frac{\beta}{\gamma}\sqrt{v+\epsilon} - m \right\rfloor}_{b}\right) \tag{21}$$

$$= \text{sign}(\gamma)\,\text{sign}\left(a + b\right) \tag{22}$$

Note that we have used $\text{sign}(0) = b_{\text{det}}(0) = +1$ here, as we have used everywhere in order to use $\text{sign}$ as a binarization function.

## C  EXPERIMENTAL DETAILS

### C.1  MNIST

The MNIST dataset consists of of 60K training and 10K test 28×28 grayscale handwritten digit images, divided over 10 classes. The images are pre-processed by subtracting the global pixel mean and dividing by the global pixel standard deviation. No other form of pre-processing or data augmentation is used. For MNIST, we use the following architecture:

$$32C3 - MP2 - 64C3 - MP2 - 512FC - SM10$$

where $X$C3 denotes a binary convolutional layer using $3 \times 3$ filters and $X$ output channels, $Y$FC denotes a fully connected layer with $Y$ output neurons, SM10 denotes a softmax layer with 10 outputs, and MP2 denotes $2 \times 2$ (stochastic) max pooling with stride 2. Note that if a convolutional layer is followed by a max pooling layer, the binarization is only performed after max pooling. All layers are followed by (stochastic) batch normalization and binarization of the activations. We use a batchsize of 128 and an initial learning rate of $10^{-2}$ Results are reported in Table 1.

### C.2  CIFAR-10

The CIFAR-10 (Krizhevsky & Hinton, 2009) dataset consists of 50K training and 10K test $32 \times 32$ RGB images divided over 10 classes. The last 5,000 images from the training set are used as validation set. Tthe images are only pre-processed by subtracting the channel-wise mean and dividing by the standard deviation. We use the following architecture for our CIFAR-10 experiment (following Shayer et al. (2018)):

$$2 \times 128C3 - MP2 - 2 \times 256C3 - MP2 - 2 \times 512C3 - MP2 - 1024FC - SM10$$

where we use the same notation as in the previous section. The Binarized Neural Network baseline uses the same architecture, except for one extra 1024 neuron fully connected layer. During training, the training set is augmented using random 0px to 4px translations and random horizontal fl Results are reported in Table 1.

