# OpenReview forum: "Probabilistic Binary Neural Networks"
_ICLR.cc/2019/Conference_

### Official Review · AnonReviewer1 · 2018-11-02
**A vaguely described idea without enough improvements**

**Rating:** 3
**Confidence:** 3

**Review:**

## Summary

This work presents a probabilistic training method for binary Neural Network with stochastic versions of Batch Normalization and max pooling. By sampling from the weight distribution an ensemble of Binary Neural Networks could further improve the performance. In the experimental section, the authors compare proposed PBNet with Binarized NN (Hubara et al., 2016) in two image datasets (MNIST and CIFAR10).

In general, the paper was written in poor quality and without enough details. The idea behind the paper is not novel. Stochastic binarization and the (local) reparametrization trick were used to training binary (quantized) neural networks in previous works. The empirical results are not significant.

## Detail comments

Issues with the training algorithm of stochastic neural network
The authors did not give details of the training method and vaguely mentioned that the variational optimization framework (Staines & Barber, 2012). I do not understand equation 1. Since B is binary, the left part of equation 2 is a combination optimization problem. If B is sampled during the training, the gradient would suffer from high variance.

Issues with propagating distributions throughout the network
Equation 3 is based on the assumption of that the activations are random variables from Bernoulli distribution. In equation 4, the activations of the current layer become random variables from Gaussian distribution. How the activations to further propagate?
Issues with ternary Neural Networks in section 2.4
For a ternary NN, the weight will be from a multinomial distribution, I think it will break the assumption used by equation 3.

Issues with empirical evidences
Since the activations are sampled in PBNET-S, a more appropriate baseline should be BNN with stochastic binarization (Hubara et al., 2016) which achieved 89.85% accuracy on CIFAR-10. It means that the proposed methods did not show any significant improvements. By the way BNN with stochastic binarization (Hubara et al., 2016) can also allow for ensemble predictions to improve performance.

---

> ### Author Response · Authors · 2018-11-26
> **Response to reviewer 1**
>
> Dear reviewer,
>
> We have tried to address the questions/remarks raised in the review. Moreover, we have updated the writing in the paper and hope the presentation is now easier to follow. Our response follows the structure of the original review such that it is easy to refer back to the original remarks.
>
> # On the general remarks
> We agree that the local reparametrization trick has been used before in order to train binary (or quantized) neural networks, however, we binarize both the weights and activations. Moreover, we propagate the activation distribution throughout the network/layer in order to backpropagate through binarization functions. By doing so, the gradient of the binarization function with respect to the parameters of the pre-activation distribution exists and can easily be computed using standard tools. Although our method doesn’t achieve better performance when compared to the Binarized Neural Networks by Hubara et al., the performance is on par. For this reason, we also do not make claims about outperforming existing methods, however, we do argue that our stochastic training method has various favorable properties, i.e., we obtain a distribution over binary network parameters that allow for any-time ensembles without retraining anything, it allows for more complex network architectures than earlier work (on probabilistic networks), and it is easily implemented in existing deep learning frameworks. Moreover, the probabilistic approach allows for straightforward inclusion of priors on the weights and/or activations which can help to impose more structure on the Binary Neural Network (e.g., sparsity priors) that can lead to even more efficient networks.
>
> ## Response to “Issues with the training algorithm of stochastic neural network”:
> We indeed didn’t elaborate in much detail on the training method because in many aspects our training method follows the standard approach in the current literature. Since we ensured that all the operations in the PBNet(-S) are differentiable (w.r.t. the parameters of the input distributions for most of the operations), we can train the PBNet as any other network (and thus leverage existing Deep Learning frameworks). For more specifics, see algorithm 1, which outlines the forward pass for a single layer.
>
> ## Response to equation 1 being unclear
> Equation 1 states the upper bound on the training objective. Our actual objective is to obtain the binary weights that minimize the loss as states on the left-hand side. This is indeed a combinatorial problem. As such we make use of the variational optimization framework (Staines & Barber, 2012) in order to obtain an upper bound on the training objective. I.e., we introduce a distribution over the binary parameters of the network and instead of optimizing the binary parameters directly, we optimize the parameters of the binary distributions. As pointed out by the reviewer, optimizing this upper bound may result in high variance on the gradients, but we deal with this in the following way:
>
> - For PBNet, we never sample weights but instead propagate the variance throughout the network, i.e., the forward pass is deterministic and we don’t suffer from high variance on the gradients
> - For PBNet-S, we leverage the local reparametrization trick, which is known to have lower gradient variance compared to simply sampling the weights during training. However, instead of sampling the (pre-)activations directly after computing the linear operation of a layer, we sample the activations at the very last operation of the layer. Moreover, the gradient variance is related to the variance of the weight distribution. For this reason, we initialize the parameters from a pre-trained network and specifically initialize the weight distribution q(B) to have low variance (compared to a random initialization).
>
> Following this, we empirically find no issues with training these networks.

---

> > ### Author Response · Authors · 2018-11-26
> > **Response to reviewer 1 cont'd**
> >
> > ## Response to “Issues with propagating distributions throughout the network “
> > The activations (and the weights for that matter) are distributed according to a scaled and translated Bernoulli distribution--which we have called the Binary distribution of ease of notation. Any inner product between two Binary distributed vectors is distributed according to a Poisson Binomial distribution. Given the fact that computing the CDF of a Poisson binomial is not straightforward [1] and that the CDF is well-approximated by the CDF of a Gaussian (under some assumptions), we approximate the distribution of the pre-activation by a Gaussian distribution. Since the actual activations of a layer are obtained by applying a binarization (activation) function to the pre-activation, we again obtain a binary distribution as activations (for which we obtain the parameters by evaluating the CDF of the Gaussian distribution). As such, for the PBNet, the input to a layer is a Binary distribution (i.e., the parameters thereof) and the output is also a Binary distribution (i.e., the parameters thereof), which allows us to stack multiple layers in a Neural Network.
> >
> > ## Response to “Issues with ternary Neural Networks “
> > We obtain a Ternary Neural Network as a post-processing step, i.e., after training, we obtain a binary distribution. We only claim that we can--as a post-processing step--set some of the weights in a BNN to zero to obtain a Ternary Neural Network. I.e., we remove the noisy weights from the network by setting them to zero. This can either be interpreted as a Ternary Neural Network or a  Sparse Binary Neural Network. In short, we never train a Ternary Network, but obtain one after training a BNN, and observe that the results improve slightly by using this post-processing step.
> >
> > ## Response on “Issues with empirical evidences “
> > We indeed do not improve over earlier work, however, our results are competitive  and our method has favorable properties compared to earlier work. For this we refer back to the first paragraph of this response. Moreover, we are aware of the results reported by Hubara et al. (2016) using stochastic activations. We chose to compare to the BNN using deterministic activiations since we were unable to reproduce the results presented on stochastic activations. Furthermore, we like to point out that the architecture used in our paper uses one fully connected layer less (following Shayer et al. (2018)), and thus has less parameters.
> >
> > In (Hubara et al., 2016), the binary weights are trained using full precision shadow weights and a deterministic binarization function (only stochastic binarization of the activations is considered). As such, after training, there is only a point estimate of the binary weights. Although it is, of course, possible to train multiple networks to obtain an ensemble, in our work, we train a single PBNet-S once and obtain multiple network instantiations from the binary weight distribution in order to create an ensemble. As such, it is possible to perform any-time ensemble predictions (without any extra cost during training), which improve both accuracy and uncertainty estimates as is shown in Table 1 and Figure 3a in our paper.
> >
> > [1] Hong, Yili. "On computing the distribution function for the Poisson binomial distribution." Computational Statistics & Data Analysis 59 (2013): 41-51.

---

> > > ### Comment · AnonReviewer1 · 2018-11-29
> > > **Thank you for the discussion**
> > >
> > > I still do not understand the training method. Equation 1 and Algorithm 1 (the  forward pass) are not enough. Please describe also the backward and the update. Please explain why this upper bound and how about the tightness.
> > >
> > > After reading the response,  I will keep my rating because "the idea behind the paper is not novel and  the empirical results are not significant."

---

> > > > ### Author Response · Authors · 2018-12-03
> > > > **thank you and some clarification**
> > > >
> > > > Thank you for taking the time to respond to our rebuttal. Since not everything is clear yet, we will take the time to give some clarifications on the points raised.
> > > >
> > > > # On the training algorithm:
> > > > We noticed that one small detail is missing from our paper, i.e., the parameters θ are constrained to lie in [-1, 1], for this reason, we update a set of unconstrained parameters Ψ and obtain θ = tanh(Ψ). As such, Ψ can be updated without constraints.
> > > >
> > > > we explicitly made sure that all operations in the PBNet(-S) are ‘auto-diffable’ both with respect to weight and input. For all layers, the inputs to the layers are the parameters (i.e., sufficient statistics) of the binary distributions that describe the activations. Note that the binary distribution includes deterministic activations of either -1 or +1 as a corner case, which also allows modeling observed, sampled activations as encountered in PBNet-S. All operations we apply on these sufficient statistics are differentiable (and can be computed using auto-diff implementations). Hence, the forward pass (which is specified for a single layer in algorithm 1) completely defines the backward pass.
> > > >
> > > > The update of Ψ is performed using Adam (as stated in our paper). Since we can backpropagate throughout the whole model, the gradient for Ψ is easily obtained using any standard backpropagation/auto-diff implementation, such as available in PyTorch. This training method is very similar to various other works that optimize an ELBO using (local) reparametrization (such as [1]), only instead of optimizing the ELBO, we optimize the expected (binary) cross entropy (similar to [2]).
> > > >
> > > > We hope this clarifies the backward pass and update step. If not, maybe the reviewer can kindly point out the source of the ambiguities? Of course, we will update the paper to include the discussion above. Moreover, we are also planning to release the code for our experiments.
> > > >
> > > > # On the bound:
> > > > We use this specific bound as it is a differentiable bound on the minimum of the non-differentiable objective function L(B). Thus, it allows us to train a Neural Network with discrete weights using gradient-based methods. On the tightness of the bound, we quote from Staines & Barber (2012): "This bound can be trivially made tight provided the distribution q(B|θ) is flexible enough to allow all its mass to be placed in the optimal state B’ = argmax_B L(B).” were we have changed the variable names to match our case. It is easy to see that in our case all the mass can be placed on the optimal state as this is the case where θ = B’.
> > > >
> > > > # On the lack of novelty:
> > > > To our knowledge, our paper introduces various novel aspects with respect to (probabilistic) Binary Neural Networks, as pointed out in our earlier responses. We would like to kindly ask the reviewer if we missed any references that we should compare the novelty of our work to?
> > > >
> > > > # On the significance of the experiments:
> > > > We like to point out that the main goal of our paper was to introduce a novel method for training Binary Neural Networks. In the experiments presented in the paper, we investigate if our method is on par with an existing method in which an equivalent Binary Neural Network is obtained. We would like to ask how we should extend our experiments in order for the reviewer to consider them significant?
> > > >
> > > > [1] Diederik P Kingma, Tim Salimans, and Max Welling. Variational dropout and the local reparameteri-zation trick. In Advances in Neural Information Processing Systems, pp. 2575–2583, 2015
> > > > [2] Oran  Shayer,  Dan  Levi,  and  Ethan  Fetaya. Learning  discrete  weights  using  the  local  repa-rameterization trick.  In International Conference on Learning Representations,  2018.

---

### Official Review · AnonReviewer2 · 2018-11-03
**The investigated problem is interesting, but methods used to handle binary networks are not impressive**

**Rating:** 5
**Confidence:** 2

**Review:**

To reduce the deep neural networks' reliance on memory and high power consumption, the paper proposed a kind of probabilistic neural networks with both binary hidden node and binary weights. The paper presents a probabilistic way to binarize the activation and weight values. Also, it proposed a random version of batch-normalization and max-pooling.

The binarization of hidden node is achieved through stochastic sampling according to the sign of the stochastic pre-activation values. The weight binarization is analogously done by sampling from a binary distribution. There is no too much new here, and is a standard way to obtain binary values probabilistically.

The paper said in the introduction that the binary model will be trained with the re-parameterization trick, through either propagating the distributions or the samples from concrete distribution. But I am still not very clear how this training process is done, especially for the training of weight parameters.

Overall, the problem investigated in this paper is very interesting and is of practical importance, the experimental results are preliminary but encouraging. But all the techniques used in this paper to binarize neural networks are standard, and no too much new here.

---

> ### Author Response · Authors · 2018-11-26
> **Response to reviewer 2**
>
> Dear reviewer,
>
> We want to clarify that we never sample from the binary weight distributions directly. We only sample the activations (i.e., the output of a layer) in some of our experiments. Moreover, it is technically correct that the stochastic binary activations are related to the sign of the stochastic pre-activations, however, we’d like to stress that the stochastic binary activation distributions are obtained from the full density of the pre-activation distribution by applying the sign function to the random variables.
>
> ## On the contributions of our paper
> First, we’d like to clarify that our paper is not only about binary neural networks, but also about probabilistc neural networks. Interestingly, by restricting ourselves to binary random variables, we get a closed-form forward pass. Although this has been explored before (e.g., Soudry et al. (2014) or Baldassi et al. (2018)), we also introduce stochastic versions of pooling and batch normalization. As a result, our method allows for more ‘modern’ or complicated probabilistic architectures. Other than that, we summarize the contributions of our paper below:
>
> - We propose a method for training Binary Neural Networks without sampling or gradient approximations (PBNet);
> - We propose a training procedure for Binary Neural Networks that has various favorable properties. E.g., the probabilistic formulation allows for any-time ensemble predictions, it allows for more complex network architectures than earlier work (on probabilistic networks), and it is easily implemented in existing deep learning frameworks;
> - We propose random/stochastic versions (as summarized in the review) of max-pooling and batch normalization, which are required in order to propagate distributions through the layer and/or network (e.g., our abblation results show that batch normalization is required and thus we need a stochastic version of batch normalization);
> - We emprically show that sampling during training is crucial for obtaining a performant deterministic Binary Neural Network, but not required when one is interested in a stochastic binary neural network (see the propagate columns in table 1).
>
> ## On the use of the (local) reparametrization trick
> The local reparametrization trick, as introduced in (Kingma et al., 2015), translates noises from global parameters to local parameters. Specifically, in (Kingma et al., 2015), instead of sampling weights from the Gaussian weight distributions, the uncertainty (or variance) of the weights is propagated to the (pre-)activations. The pre-activations are subsequently sampled from the pre-activation distributions using the reparametrization trick. In (Kingma et al., 2015), the weights are Gaussian and the activations are scalars, hence the pre-activations are Gaussian too, however, it is also possible to obtain Gaussian approximations to the pre-activation distributions by following the Lyapunov Central Limit Theorem. If we would simply follow the reparametrization trick, we would sample from this distribution at this point (as was done by Shayer et al. (2018)). However, based on the observation that we can back-propagate through binarization (or step-wise) functions when applied to random variables (if the CDF of the corresponding distribution is differentiable), we choose to apply the binarization function (i.e., non-linearity) before sampling. In the case of PBNet-S we then sample the binary activation using the concrete relaxation of the Bernoulli distribution (using the reparametrization trick). In the case of PBNet, we do not sample the activation but propagate the binary activation distribution to the next layer. As such, technically, PBNet does not make use of the local-reparametrization trick. We have updated the introduction to make this distinction more clear.

---

### Official Review · AnonReviewer3 · 2018-11-04
**Overall score 6**

**Rating:** 6
**Confidence:** 4

**Review:**

1. This paper presents a novel approach for training efficient binary networks with both binary weights and binary activations and verifies the capability of the proposed method on benchmark. In addition, related bach normalization and pooling layers are also improved.

2. How long do we need for training BNNs using the proposed method?

3. The paper only compares their performance with only one related work (Hubara et al., 2016), which is somewhat insufficient.

4. Another only concern is that whether the proposed method can be well applied on large scale datasets such as ImageNet.

---

> ### Author Response · Authors · 2018-11-26
> **Response to reviewer 3**
>
> ## On training time
> Figure 3c in our paper shows a plot of the validation loss. This plot shows that training on cifar-10 converges in approximately 50 epochs. Of course, during training we do have to perform two convolutions (one for the mean and one for the variance of the Gaussian approximation), which may result in a longer wall clock time, if the hardware does not permit parallelization of these two operations.
>
> ## On the comparison to other methods
> We compare to Hubara et al.’s Binarized Neural Networks as it is the natural alternative to train Binary Neural Networks. The binary network that one obtains after training user our method is equivalent to the Binary Network obtained using the method by Hubara et al. Moreover, in both approaches no new computations are introduced (such as, for example, in XNOR-net (Rastegari et al., 2016) or ABC-net (Lin et al., 2017)). For this reason, we believe it is fair to compare our method to that of Hubara et al., however, we do agree that is may be interesting to explore the probabilistic counterparts of, for example, XNOR-Net and ABC-Net in future work.
>
> # On large scale datasets
> Although we agree that experiments on larger scale datasets are valuable, we also believe that the results currently presented in our paper are already valuable. We did perform some preliminary experiments on larger scale datasets and found evidence that we can scale to larger datasets, however, some careful tuning may be required, as is often the case for binary neural networks.

---

### Meta-Review · Area_Chair1 · 2018-12-12
**Limited novelty and non-impressive experimental results**

**Confidence:** 4
**Recommendation:** Reject

**Metareview:**

The paper proposes a probabilistic training method for binary Neural Network with stochastic versions of Batch Normalization and max pooling.

The reviewers and AC note the following potential weaknesses: (1) limited novelty and (2) preliminary experimental results.

AC thinks the proposed method has potential and is interesting, but decided that the authors need more works to publish.